# Using Redox Proteomics to Gain New Insights into Neurodegenerative Disease and Protein Modification

**DOI:** 10.3390/antiox13010127

**Published:** 2024-01-20

**Authors:** Paula Cadenas-Garrido, Ailén Schonvandt-Alarcos, Lourdes Herrera-Quintana, Héctor Vázquez-Lorente, Alicia Santamaría-Quiles, Jon Ruiz de Francisco, Marina Moya-Escudero, David Martín-Oliva, Sandra M. Martín-Guerrero, César Rodríguez-Santana, Jerónimo Aragón-Vela, Julio Plaza-Diaz

**Affiliations:** 1Research and Advances in Molecular and Cellular Immunology, Center of Biomedical Research, University of Granada, Avda, del Conocimiento s/n, 18016 Armilla, Spain; paula.cadenas.garrido@gmail.com (P.C.-G.); ailen.schonvandt@gmail.com (A.S.-A.); aliciasantamariaquiles@gmail.com (A.S.-Q.); jonruizdef@gmail.com (J.R.d.F.); marinamoyaescudero@gmail.com (M.M.-E.); 2Department of Physiology, Schools of Pharmacy and Medicine, University of Granada, 18071 Granada, Spain; lourdesherrera@ugr.es (L.H.-Q.); hectorvazquez@ugr.es (H.V.-L.); cerosan.18@gmail.com (C.R.-S.); 3Biomedical Research Center, Health Sciences Technology Park, University of Granada, 18016 Granada, Spain; 4Department of Cell Biology, Faculty of Science, University of Granada, 18071 Granada, Spain; dmoliva@ugr.es; 5Department of Basic and Clinical Neuroscience, Institute of Psychiatry, Psychology and Neuroscience, King’s College London, London SE5 9RT, UK; 6Department of Health Sciences, Area of Physiology, Building B3, Campus s/n “Las Lagunillas”, University of Jaén, 23071 Jaén, Spain; 7Children’s Hospital of Eastern Ontario Research Institute, Ottawa, ON K1H 8L1, Canada; 8Department of Biochemistry and Molecular Biology II, School of Pharmacy, University of Granada, 18071 Granada, Spain; 9Instituto de Investigación Biosanitaria IBS, Complejo Hospitalario Universitario de Granada, 18071 Granada, Spain

**Keywords:** redox proteomics, oxidative stress, protein, health, neurodegenerative diseases

## Abstract

Antioxidant defenses in biological systems ensure redox homeostasis, regulating baseline levels of reactive oxygen and nitrogen species (ROS and RNS). Oxidative stress (OS), characterized by a lack of antioxidant defenses or an elevation in ROS and RNS, may cause a modification of biomolecules, ROS being primarily absorbed by proteins. As a result of both genome and environment interactions, proteomics provides complete information about a cell’s proteome, which changes continuously. Besides measuring protein expression levels, proteomics can also be used to identify protein modifications, localizations, the effects of added agents, and the interactions between proteins. Several oxidative processes are frequently used to modify proteins post-translationally, including carbonylation, oxidation of amino acid side chains, glycation, or lipid peroxidation, which produces highly reactive alkenals. Reactive alkenals, such as 4-hydroxy-2-nonenal, are added to cysteine (Cys), lysine (Lys), or histidine (His) residues by a Michael addition, and tyrosine (Tyr) residues are nitrated and Cys residues are nitrosylated by a Michael addition. Oxidative and nitrosative stress have been implicated in many neurodegenerative diseases as a result of oxidative damage to the brain, which may be especially vulnerable due to the large consumption of dioxygen. Therefore, the current methods applied for the detection, identification, and quantification in redox proteomics are of great interest. This review describes the main protein modifications classified as chemical reactions. Finally, we discuss the importance of redox proteomics to health and describe the analytical methods used in redox proteomics.

## 1. Introduction

The evolution of aerobic metabolism occurred approximately 2.2 billion years ago, which allowed organisms to have more effective energy metabolisms and develop into new complex multicellular organisms [1]. In addition, the reduction of molecular oxygen into chemically reactive derivatives, collectively known as reactive oxygen species (ROS), can also be highly toxic to biological systems. It has been reported that some components of cells, such as deoxyribonucleic acid (DNA), membrane lipids, and proteins, react with diffusion-limited rate constants at diffusion-limited rates [2,3]. In response to ROS, cells have evolved extensive defense mechanisms, including (i) enzyme activities (such as catalase (CAT)), (ii) antioxidant response genes, (iii) antioxidant molecules (such as glutathione (GSH)), and (iv) thioredoxin systems [4].

Antioxidant defenses in biological systems maintain a state of redox homeostasis in which background ROS levels can be effectively managed by antioxidant defenses. The state of oxidative stress (OS) occurs when ROS levels exceed existing antioxidant defenses or when such defenses decline (mainly as a result of aging) [5]. In addition, it has become increasingly evident that ROS and interrelated reactive nitrogen species (RNS) play an important role in biological signaling, even under conditions of substress [6,7]. ROS are absorbed primarily by proteins, which can confound the proteome via changing individual proteins levels through mechanisms such as changes in the gene expression and modified degradation of altered proteins. In addition, ROS can alter the covalent protein’s structure, which results in pointed modifications to protein function [2,4].

The field of redox proteomics consists of characterizing the oxidation of proteins and determining the magnitude and site of oxidative modifications that have been introduced into a proteome of interest [8,9,10]. The molecular pathways involved in protein oxidation and human disease can be identified once oxidatively modified proteins are detected [11,12,13]. Proteomics offers the advantage of obtaining complete information about a cell’s proteome, which is constantly changing as a result of both its genome and environment interactions [14,15]. Together with measuring the levels of protein expression, proteomics is capable of determining both protein modifications and localizations, the effects of added agents, and interactions between proteins [16].

Cysteine (Cys), methionine (Met), and selenocysteine are capable of undergoing reversible redox reactions. This is necessary for the dynamic regulation of the function and structure of proteins [17]. It is also possible for them to serve as reliable molecular indicators of irreversible oxidative damage to a variety of amino acids in diverse pathophysiological circumstances. This will add to the establishment of a link among the pathological trademarks of disease and irregular protein structure/function [17].

Biomolecules are continually exposed to oxidative impairment to cellular components, and that is progressively accepted as one of the most critical pathophysiological events underlying disease and aging [12]. Tobacco smoke, pollution, alcohol, and certain drugs are exogenous sources of ROS and other oxidizing species [12]. The first attempt at identifying OS-induced protein glutathionylation in T cell blasts using redox proteomics was performed in 2002 [9]. Also in the same year, redox proteomics was described in relation to oxidized hippocampal proteins in Alzheimer’s disease patients [18]. Several reports have been conducted since then using redox proteomics to understand how the structure of protein disturbs both health and disease [19,20,21]. It is important to describe the main methods used to make the modifications in order to understand their key features.

In this review, we describe the main protein modifications that are categorized as chemical reactions. We also discuss the importance of redox proteomics to health, and finally, we describe the analytical methods in redox proteomics. The main aspects of the present review are graphically represented in Figure 1.

## 2. Protein Modifications Evaluated in Redox Proteomics

Proteins are highly sensitive to OS caused by ROS and RNS, leading to different modifications that may be reversible, usually at Cys residues; these modifications function as a protective mechanism against irreversible damage and as a modulator of protein function (redox regulation) [22]. On the other hand, ROS and RNS also produce unalterable modifications (e.g., di-Tyr formation, cross-linking between proteins, carbonylation of Lys and Arg) that may result in permanent functional loss and damaged proteins’ degradation [23]. Furthermore, they may accumulate into cytoplasmic inclusions over time, as has been observed in some aging-related diseases [24].

Additionally, post-translational modifications of proteins by oxidative means are a frequent phenomenon, including (i) carbonylation that results from the free radical scission of the primary peptide chain, amino acid side chains oxidation, glycation of the products of reactions between proteins and reducing sugars, or the lipid peroxidation products that produce highly reactive alkenals [25,26,27]; (ii) covalent adducts that are formed when 4-hydroxy-2-nonenal or other reactive alkenals are Michael added to Cys, Lys, or His residues [26,27]; and (iii) nitration of tyrosine (Tyr) residues and nitrosylation of Cys residues [28].

### 2.1. Oxidative/Nitrosative Modification of Cysteine

Cys residues are finely sensitive to cellular redox, prime targets for oxidative modifications. Many proteins are controlled by thiol functionality [21]. Those proteins that are redox-sensitive contain at least one Cys residue, which is surrounded by amino acids with basic properties in the tertiary or quaternary structure, lowering their pKa values—this pKa influencing the susceptibility to oxidation by many ROS/RNS—and leading to deprotonation. Moreover, a number of factors may eventually reduce the pKa of Cys, including helix-dipole effects and hydrogen bonds with serine or histidine (His) residues [29].

Protein thiol groups may exhibit different modifications and the sulfur atom of the Cys thiol moiety undergoes three oxidation states. These states may be altered in response to a variety of stimuli and changes in cellular redox states [21]. For instance, S-glutathionylated species are produced in mammalian cells when significant amounts of GSH are consumed through an oxidative process—S-glutathionylation (SSG) being considered a mechanism for storing GSH under stress and protecting against irreversible protein thiol oxidation—[30,31]. On the other hand, when Cys are mildly oxidized, they may produce sulfenic acid (P-SOH), protein-mixed disulfides with low-molecular-weight thiols (e.g., GSH), intermolecular disulfides, and S-nitrosothiols. Cys can cycle between the oxidized and reduced states of the molecule as a consequence of these reversible modifications. Likewise, in reactions with other reversible oxidative modifications of sulfhydryl groups (SHs), sulfenic acid is frequently an intermediary in the synthesis of sulfonic and/or sulfinic acids [32]. A strong oxidative insult to sulfonic (P-SO_3_H) and sulfinic acids (P-SO_2_H) can irreversibly oxidize Cys residues, which cannot normally be undone by metabolic processes and may result in a protein’s loss of function. In addition, it was demonstrated that sulfinic inactive forms of peroxiredoxin I are rapidly converted to catalytically active thiol forms [33]. In summary, the oxidative modifications of Cys residues in proteins may cause irreversible damage to the catalytic site. Moreover, it may work as a switch for regulating protein structure or activity [34].

On the other hand, it must be noted that Cystine (CySS)—the dimeric and the oxidized form of Cys—is the predominant form in the extracellular space, whereas Cys is more prevalent at the intracellular level due to the reductive environment of cells [35]. The redox state of plasma is largely determined by the redox state of Cys/CySS, which shifts towards a more oxidizing state in the course of aging [36]. Thus, aging leads to a decline in redox buffering capacity along with a decrease in GSH and Cys levels [37,38].

A number of post-translational modifications are also carried out on Cys, which modulate many physiological processes. There is increasing evidence that redox-modulated events have important functions not only in peripheral tissues, but also in the brain, where Cys disposition plays a significant role. Indeed, several neurodegenerative disorders are associated with dysregulated Cys metabolism [39]. In this regard, the cystine–glutamate antiporter (system X_c_^−^) represents an intriguing target in attempts to understand the pathological states of the central nervous system (CNS) [40].

### 2.2. Oxidative/Nitrosative Modification of Tyrosine

Specific Tyr derivatives generated by oxidation have been observed in different conditions, such as myeloperoxidase-mediated reaction pathways with HOCl (Cl-Tyr, di-Cl-Tyr, and 3,5-dichlorotyrosine) [41,42]. A number of reactions are eosinophil peroxidase-catalyzed by hypobromous acid (Br-Tyr, 3-bromotyrosine, and 3,5-dibromotyrosine, di-Br-Tyr) [22], RNS-catalyzed pathways (NO_2_-Tyr, di-Tyr, phenylalanine, and quinone), and free radical pathways (o-Tyr, m-Tyr, and di-Tyr) [22].

Tyr nitration is one of the most remarkable modifications, being a selective process which may alter protein function, Tyr being incapable of performing electron transfer reactions and maintaining protein conformation as a result of bulky groups inserted onto aromatic rings which lower phenolic group pKa [43,44]. Several studies have suggested that peroxynitrite—one of the major oxidants formed in vivo under intense OS—is the primary intermediary of these modifications [45,46]. However, it must be noted that other processes and molecules are involved in Tyr nitration, such as inflammatory oxidative reactions, which can also trigger facile Tyr nitration pathways (e.g., myeloperoxidase and other metalloproteins that possess peroxidase activity are efficient at catalyzing Tyr nitration in the vascular compartment) [47,48]. Moreover, there are several substrates necessary for Tyr nitration (hydroperoxides, NO, and NO_2_), but when these substrates become abundant and accessible for myeloperoxidase-dependent nitration [43,49], multiple mechanisms are likely to contribute to the formation of tissue NO_2_-Tyr. In general, RNS have been shown to produce mainly NO_2_-Tyr but are also capable of simultaneously form di-Tyr and oxidizing even more RNS-susceptible protein targets (such as FeS, Cys, or ZnS complexes) [49]. On the other hand, it must be noted that Tyr nitration may result in either increased or decreased function [50], and this nitration can be reduced or removed through either nonenzymatic or enzymatic mechanisms [51].

Post-translational proteins are therefore a sign of nitrosative/oxidative injury and are commonly associated with modified protein function in inflammatory conditions [52,53]. Considering that NO_2_-Tyr formation can be reversed [51], Tyr nitration may be more than just a marker of RNS formation due to its role upon disturbing protein function, as was lately confirmed for actin in both murine models and human sickle cell tissues [52].

### 2.3. Oxidative/Nitrosative Modification of Methionine

Almost all ROS/RNS are capable of degrading Met residues [54]. The generation of methionine sulfoxide (MetO) under mild oxidizing conditions can further be oxidized to methionine sulfone (MetO_2_) in more robustly oxidizing environments [55]. Moreover, in the presence of ROS/RNS, two diastereomeric forms are formed: Met-R(O) and Met-S(O). Therefore, Met oxidation has been associated with a loss of protein function (e.g., A-1 protease inhibitors, a-chymotrypsins, ribonucleases, phosphoglucomutases, actins, or peptide hormones) [22,54]. However, this phenomenon is not necessarily caused by Met oxidation [56].

The enzymatic reversibility of Met oxidation has been demonstrated. Met sulfoxide reductases (Msrs) are enzymes that catalyze the reduction of MetO in response to NADPH. Furthermore, the cell contains two different types of stereospecific Msrs: one that is specific for the S-isomer (MsrA) and one that is specific for the R-isomer (MsrB) [57,58]. Thus, it has been suggested that Met residues may act as an endogenous antioxidant defense to protect targeted proteins from irreversible oxidative damage to other essential amino acids [55]. This idea is consistent with the observation that Met oxidation is not associated with an increase in proteolytic degradation susceptibility, unlike other amino acid modifications [59].

In this line of research, it has been observed that transgenic mice with a knockout of MsrA exhibit shortened life spans and increased levels of protein carbonyl, supporting the idea that the overexpression of Msrs may confer resistance to oxidants since Met residues in proteins serve protective roles by preventing oxidative damage at other residues [55].

### 2.4. Oxidative/Nitrosative Modification of Histidine and Tryptophan

Trp and His residues are also susceptible to be oxidized, causing specific amino acid modifications, especially when these residues are close to heme or copper (Cu)(II) binding sites [22]. Upon attachment to precise metal-binding sites on proteins, Cu(I) or iron (Fe)(II) ions generate radicals, which oxidize adjacent amino acid residues. Ascorbate, mercaptane, or NADH are suitable electron donors for the metal-catalyzed oxidation of proteins, which occurs at transition-metal-interacting sites, resulting in a highly selective reaction [60].

When His is oxidized, 2-oxohistidine and 4- or 5-hydroxy-2-oxo-histidine are produced [22]. The Trp oxidation products include N-formylkynurenine, 3-OH-kynurenine, hydroxytryptophan, and kynurenine. These protein metal-catalyzed oxidations have been observed in myoglobin [61], Cu, zinc-superoxide dismutase (CuZn-SOD) [62], β-amyloid peptide [63], and recombinant prion protein [64]. Further, the degree of modification of His or Trp can be determined by proteins that are in close proximity to a general source of ROS, such as normally respiring tissues or ultraviolet radiation. For instance, it has been reported that N-formylkynurenine can be generated under normal conditions by cardiac mitochondrial proteins [65].

### 2.5. Protein Carbonylation and Glycation End Products

The carbonylation of proteins is an irreversible oxidative process. These carbonylated proteins cannot be repaired and are either degraded or accumulated as damaged or unfolded proteins [66]. Furthermore, it has been observed that the OS damage in proteins correlates perfectly with the number of carbonyl groups. Thus, protein carbonylation is considered a suitable marker of the degree of oxidative damage in proteins related to several diseases [67].

There are numerous ways to introduce carbonyl groups into a protein (aldehydes and ketones) [66]. As ketone or aldehyde derivatives are produced from amino acids such as proline (Pro), arginine (Arg), Lys, and threonine (Thr), carbonyl groups are formed on their side chains [68]. It is also possible to generate protein carbonyl groups through the oxidative cleavage of proteins, either through α-amidation or by the oxidation of glutamyl groups on the side chains, leading to the formation of peptides with a-ketoacyl derivatives that block the amino acid at the N-terminus [68]. A protein can also be carbonylated through the reaction of 2-propenals (acrolein), 4-hydroxynonenals (4-HNE), and malondialdehydes (MDA). In this reaction, reactive aldehydes are added via a Michael addition to Cys, His, or Lys residues, which results in the incorporation of the aldehyde/carbonyl group into the peptide chain [69].

In addition to reactive carbonyl groups, secondary reactions involving reducing sugars or their oxidation products (glycation and glycoxidation) can also produce reactive carbonyl groups (ketoamines, ketoaldehydes, and deoxyosones) [66]. By altering the conformation of the polypeptide chain, these carbonyl moieties may partially or completely inactivate a variety of proteins.

As a result of nonenzymatic reactions between reduced sugars and proteins, lipids, or nucleic acids, advanced glycation end products (AGEs) are generated. A variety of biological responses are induced by AGEs when they bind to one or more of their multiple receptors (RAGE) that are found on a variety of types of cells [70].

An Amadori product is formed after some rearrangement of the Schiff base, which is then spontaneously transformed at physiological pH and room temperature into a reversible Schiff base. Depending on the pH, these Amadori products may also undergo enolization reactions, resulting in 1,2-dicarbonyls, which may later be dehydrated to yield furfural derivatives [71]. A variety of ROS have been reported as a result of AGEs activating NADPH oxidases [72,73], the mitochondrial respiratory chain, microsomal enzymes, xanthine oxidase and arachidonic acid pathways [74,75,76] by interacting with their receptors. ROS are generated when AGEs interact with RAGE, as there is considerable evidence to support this claim. Figure 2 summarizes the main protein modifications described in this section.

## 3. Redox Proteomics in Neurodegenerative Diseases

It has been suggested that OS is implicated in many neurodegenerative diseases as a result of oxidative damage to the brain, which is particularly vulnerable to oxidative damage due to its high consumption of dioxygen [77]. Moreover, age-related neurodegenerative diseases are linked to altered metabolic pathways, with particular disturbances in protein metabolism, the appearance of abnormal protein aggregates in some of these diseases being usual. Thus, the study of redox proteomics may be an enormous source of information, which can be combined with additional data of epigenomics, genomics, metabolomics and/or lipidomics, thus providing new details on the molecular structure and function (or dysfunction) of complex systems involved in neurodegenerative diseases [78]. Specifically, we examined Parkinson’s disease (PD), Alzheimer’s disease (AD), Huntington’s disease (HD), multiple sclerosis (MS), and amyotrophic lateral sclerosis (ALS). The main findings on redox proteomics are summarized in Table 1.

### 3.1. Parkinson’s Disease

PD is an age-related neurodegenerative disorder characterized by the progressive death of selected dopaminergic neurons, affecting about 6 million people and increasing over the years [79]. The hallmark of this pathology is the formation of Lewy bodies, which are misfolded and fibrillary forms of α-synuclein (α-syn) in surviving neurons [80]. This aberrant α-syn accumulation resulting from proteasome dysfunction—an abnormal immunoproteasome assembly being a possible key contributor—is considered as a prominent factor to initiate and aggravate the neurodegeneration [81]. In fact, the aggregation of α-syn monomers has been shown to generate beta-sheet-rich oligomers which cause mitochondria impairments and oxidation events [82]. On the other hand, mitochondrial dysfunction has been detected prior to neuronal loss and α-syn fibril deposition, suggesting that mitochondrial dysfunction is one of the key drivers of early disease [83].

Only 5% of PD cases are associated with mutations in a few genes (i.e., the early onset or familial parkinsonism) whereas most cases are sporadic [84].

The most common mutations in familial parkinsonism are those that affect the oxidation and nitration of α-syn, together with a missense mutation in the α-syn gene (*SNCA*) producing the amino acid substitution A53T in the protein [85], *LRRK2* genes (causing autosomal dominantly inherited disease) or *PINK1* and parkin genes (which are related to autosomal recessively inherited forms of PD) [86,87]. In general, the proteins that are encoded by these genes seem to have a vital part in the normal function of mitochondria and their recycling in case of damage, so their mutation may lead to mitochondrial dysfunction and PD development [87]. For instance, parkin—a principally cytosolic Cys-rich protein with diverse cellular functions such as the maintenance of mitochondrial integrity—is associated with a protective effect against OS, and the loss of these complementary redox effects (i.e., the capacity of oxidative modifications of parkin Cys) may augment OS during ageing [88].

Regarding sporadic PD, it has been observed that mitochondria located in the substantia nigra are dysfunctional because of a decrease in complex I activity, which causes a reduction in ATP and an increase in ROS production [86]. Despite the highly complex etiology of PD, all of the proposed pathomechanisms of the disease are known to have a major impact on the oxidative burden of the cell, and thus strongly influence the nonenzymatic post-translational modifications of proteins [89]. For example, ceruloplasmin—a ferroxidase present in cerebrospinal fluid (CSF)—is oxidized and deamidated in PD [90]. In this line of research, recent investigations highlight the potential role of the deregulation of the sensitive Cys proteome as a convergent pathogenic mechanism [84].

### 3.2. Alzheimer’s Disease

According to the 2019 Global Burden of Disease, 51.6 million people worldwide suffered from AD in 2019 [91], the disease being responsible for approximately 50–75% of all dementias [92]. Different biomarkers have been clinically validated to allow for a differential diagnosis of AD, such as CSF levels of total tau [93,94], the neuropathological hallmarks of this disease being the abnormal deposition of amyloid-β (Aβ) fibrils—the main component of senile plaques (SPs)—and the intracellular accumulation of neurofibrillary tangles (NFTs), composed of hyperphosphorylated tau protein (pTau) [95]. OS is implicated in the pathogenesis and progression of AD, which is a source of OS that, when associated with Aβ-peptide, may lead to lipid peroxidation [96]. Biomolecules are oxidized primarily within neuronal membranes and their integrity is disrupted, observing augmented levels of Aβ-1-40 and 1-42 in the hippocampus and cortex, the brain areas most related to AD [97]. Moreover, mitochondrial dysfunction has also been associated with AD pathogenesis, through ROS generation by mitochondrial peptides in the presence of metal ions [98,99]. Additionally, it is common to find the presence of carbonylated proteins, historically linked with protein oxidation, particularly in the parietal cortex and hippocampus [100], and a higher degree of oxidative damage to membrane proteins than to cytoplasmic proteins in the human brain [101].

Several proteins involved in glucose metabolism and ATP synthesis (i.e., triosephosphate isomerase, fructose biphosphate aldolase, phosphoglucose mutase, enolase, glyceraldehyde phosphate dehydrogenase, and pyruvate kinase) have been shown to be inactivated in AD brains as a result of OS [102]. Furthermore, there is evidence that OS may play a function in the clearance of Aβ, which would oxidize low-density-lipoprotein receptor-related protein 1 and 2 (LRP1 and LRP2 [103]), resulting in the increase in the neurotoxic peptide Aβ in the brain. Certainly, LRP1 and LRP2 are key proteins that are directly associated with the Aβ efflux at the blood brain barrier [104,105], AD being known to reduce this efflux [106]. In the hippocampus of individuals with AD, LRP1 and LRP2 undergo oxidation. These oxidative modifications to LRP1 and LRP2 could result in structural changes, thereby impacting their capability to efflux Aβ, suggesting a potential mechanism [107,108].

AD also involves OS on the tau protein. By inducing modifications to the conformation of protein tau, 4-hydroxynonenal has been revealed to show a part in the pathogenesis of Alzheimer’s disease by fostering the formation of NFTs [109]. As well as promoting a conformational change, the nitration of protein tau may also facilitate fibril assembly. Since nitrated tau appears in NFTs before a tau inclusion has matured, it represents an early event during Alzheimer’s disease [110].

Apoε4, which is the main carrier of cholesterol in the brain, is considered to impact the level of OS since the plasma from AD apoε4 holders was more oxidized than the plasma from AD non-apoε4 holders [111,112]. There is evidence that this genotype may impact the metabolism of cholesterol and the formation of oxysterols [113].

In the brain, Fe, Zn, and Cu are the three most abundant ions [114], a strong correlation existing between these three metal ions and Aβ and/or OS. Proteins usually bind Cu, Fe, and Zn ions to control their reactivity, and most often, they serve as catalytic centers, electron transfer sites, or structural components of metalloproteins. In this context, metallothioneins (MTs) are proteins rich in Cys that binds to metals, possessing antioxidant and anti-inflammatory properties. The involvement of MTs in AD is evident as Aβ plaques are enriched with Zn, Cu, and Fe, metals likely playing a role in Aβ aggregation, thereby contributing to plaque formation and the generation of ROS [115]. Compared to AD model mice, amyloid plaques from humans gather higher levels of metals [116]. On the other hand, it must be noted that AD also affects a large number of transporters and metalloproteins, as well as metal concentrations, metal distribution, and homeostasis [117]. Moreover, it was reported that loosely bound Cu levels were greater in AD brains compared with healthy brains [118], free ions of Cu and Fe being involved in OS processes [119].

### 3.3. Huntington’s Disease

HD is an autosomal dominant inherited disease by the overexpression of huntingtin (HTT), a neuronal toxic protein [120,121]. The CNS is most influenced by the abnormal expression of HTT, which results in the most characteristic symptoms being psychiatric and motor problems [121]. While the normal HTT participates in cell survival, the mutated protein tends to aggregate and accumulate in neurons’ nuclei, causing the disease [19]. One of the proposed and more consistent hypotheses to explain the damage that is caused by these aggregates of abnormal HTT that comes from the mutation of the *HTT* gene may lead to OS and high ROS levels by disturbing mitochondrial functions [19,120].

It has been proposed that, apart from damaging mitochondrial DNA and improving ROS generation and OS (causing the previously mentioned mitochondria dysfunction), this mutated protein may also impair DNA repair by reducing APE1 accumulation in mitochondria, as this endonuclease plays a major role in repairing DNA that has been damaged due to OS in different cell localizations (including mitochondria) [121,122].

Oxidative damage has been demonstrated to play a crucial role in both the pathogenesis and progression of HD. In connection with this, an animal model expressing HD revealed a significant oxidation of six proteins. These proteins included α-enolase, γ-enolase (neuron-specific enolase), aconitase, voltage-dependent anion channel 1, heat shock protein 90, and creatine kinase [123]. Moreover, increased levels of oxidative damage, including protein carbonylation, have been identified in the brains affected by HD. Oxidative stress could potentially disrupt proteasome function, leading to an escalation in the aggregation of an N-terminal fragment of HTT and subsequent increases in cell death [124].

According to some studies, the use of certain antioxidants as therapies seems to reduce disease progression [125,126]. Moreover, the use of interference RNA or antisense oligonucleotides has been proposed as effective therapies to treat this genetic disorder, but it would be interesting to develop strategies that target the mentioned mitochondrial pathway to decrease the damages it causes [127].

### 3.4. Multiple Sclerosis

Inflammatory disorders of the CNS such as MS are multifactorial autoimmune disorders characterized by demyelination, axon loss, and neuronal death. Myelin sheaths that cover nerve fibers are attacked by the immune system, causing progressive neurodegeneration. A recent estimate from the Multiple Sclerosis International Federation indicates that the global prevalence of MS is approximately 2.8 million people [128]. There are several mechanisms contributing to the gradual accumulation of disability associated with this disease, including OS through the generation of ROS and RNS as well as mitochondrial damage [129].

It has been reported in some studies that patients with MS have higher levels of OS markers than healthy individuals. Active MS lesions, for example, are associated with profound mitochondrial protein alterations and DNA deletions in neurons [130]. Axons with disturbed transport are more likely to accumulate oxidized phospholipids than oligodendrocytes in these lesions. It has been demonstrated that mitochondrial damage is limited to the lesion area, even in the absence of demyelination, in CNS autopsies of human with MS [131]. A number of other studies have demonstrated extensive oxidative damage to proteins, lipids, and nucleotides in active demyelinated MS regions, particularly in reactive astrocytes and myelin-loaded macrophages [132]. This damage appears to be related to the extent of inflammation [133]. Scavenging activity is also increased in these lesions as a result of augmented antioxidant enzyme activity, such as CAT, heme oxygenase 1, and superoxide dismutase 1 and 2 (SOD1 and SOD2) [132], and an increase in Nrf2 transcription in infiltrating macrophages [134]. Furthermore, the blood–brain barrier dysfunction in MS patients is associated with nitric oxide metabolites that are increased in their CSF samples. These findings suggest that nitric oxide may play a detrimental role in the impairment of the blood–brain barrier in MS patients. It appears that activated microglia and macrophages orchestrate tissue injury through the release of oxidative bursts during the development and progression of MS lesions. Moreover, higher numbers of activate microglia contribute to neurotoxicity by releasing proinflammatory cytokines, reactive oxidative species, proteases, and glutamate, thus promoting lesion evolution and the progression of MS [135]. Even though a complex antioxidant response is simultaneously triggered, it is not sufficient to reverse the degeneration and apoptosis processes [136].

Elevated protein carbonylation has been observed in the postmortem white matter and grey matter tissue of MS patients [137], indicating evidence of nitrosative stress in demyelinated lesions and oxidative DNA damage in MS plaques. Furthermore, a significant portion of the plasma low-density lipoprotein (LDL) entering the parenchymal plaques of MS patients undergoes oxidative modifications within the lesion [138]. Moreover, transthyretin (TTR) is a homotetrameric protein found in the CNS that serves as the primary carrier for thyroxine 4 (T4) transportation from the blood to the CSF and facilitates remyelination following damage to the myelin sheath. Elevated levels of S-sulfhydration and S-sulfonation occur exclusively in cerebral TTR. These modifications correlate with abnormal TTR protein folding and are associated with the duration of the disease [139].

Physiological ROS signaling should be prevented from being reversed by therapeutic treatments for MS and other diseases, including redox balance. There is evidence that dimethyl fumarate (DMF) activates antioxidative pathways and increases Nrf2 expression [140]. As a result of stabilizing oligodendrocyte metabolism, DMF may preserve the integrity of myelin by providing protection against oxidant challenges [141]. Despite not fully understanding the mechanism of action of DMF in MS treatment, it has been confirmed to be a safe antioxidant agent. The use of antioxidant dietary compounds as complementary therapies can enhance the beneficial effects of DMFs by exerting similar functions and enhancing their beneficial effects.

### 3.5. Amyotrophic Lateral Sclerosis

Amyotrophic lateral sclerosis (ALS) is a multisystem neurodegenerative disease [142]. One of the main contributors to this disease is OS, as it appears to be intimately associated with a series of cellular events in motor neurons that contribute to neuronal death and degeneration [143]. In fact, postmortem analyses of neuronal tissue from patients with ALS consistently reveal oxidative damage to proteins, lipids, and DNA [144,145,146,147,148], suggesting that oxidative injury may constitute a major cellular mechanism underlying motor neuron degeneration.

In several studies, OS markers are elevated in patients with ALS compared to healthy individuals. In a study by Bogdanov et al., 8-Oxo-deoxyguanosine was found to be elevated in the CSF and urine of ALS patients in comparison to controls [149]. Simpson et al. [150] found that serum and CSF levels of 4-HNE were significantly higher in patients with ALS than those in a healthy control group, and that HNE concentrations increased over time and were positively correlated with the progression of the disease. Preclinical and clinical studies have demonstrated the potential for therapeutic strategies targeting OS. In this regard, in patients with ALS, treatment with antioxidants such as coenzyme Q_10_ [151], inosine [152], and rasagiline [153], has been shown to slow disease progression. However, to develop effective therapeutic strategies for ALS, more research is required in order to fully understand how OS contributes to the disease.

Patients with ALS exhibit potential pathogenic mechanisms, including nitrative stress and disruptions in redox regulation [154]. Three proteins (translationally controlled tumor protein (TCTP), ubiquitin carboxyl-terminal hydrolase-L1 (UCH-L1), and αB-crystallin) have been shown to exhibit elevated specific carbonyl levels in ALS. Given that oxidative modification can result in structural changes and diminished activity, the oxidative modification of these proteins might play a crucial role in the neurodegeneration observed in ALS [155].

**Table 1 antioxidants-13-00127-t001:** Findings on redox proteomics previously identified in neurodegenerative diseases.

Neurodegenerative Disease	Protein	Findings on Redox Proteomics
*Parkinson’s disease*	α-syn	Proteasome dysfunction causes aberrant α-syn fibril deposition, leading to the formation of Lewy bodies and a possible existing relationship with mitochondrial dysfunction
	Parkin	Cys-rich protein with diverse cellular functions such as the maintenance of mitochondrial integrity or protective effect against OS
	Ceruloplasmin	In PD, CSF ceruloplasmin (a ferroxidase) has been shown to suffer oxidation and deamidation processes
	Cys proteome	Cys proteome, in general, has been pointed out as a promising marker in PD and other neurodegenerative diseases
*Alzheimer’s disease*	Amyloid-β	Aβ aggregation causes abnormal deposition of Aβ fibrils (the main component of SPs)
	Tau protein	Total tau and phosphorylated tau increase in AD; NFTs are composed of hyperphosphorylated tau protein
	LRP1 and LRP2	Proteins directly associated with the Aβ efflux at the blood–brain barrier
	Carbonylated proteins	A common find, particularly in the parietal cortex and hippocampus
	Proteins involved in energy metabolism	OS cause inactivation of different proteins in AD brains (i.e., triosephosphate isomerase, fructose biphosphate aldolase, phosphoglucose mutase, enolase, glyceraldehyde phosphate dehydrogenase, and pyruvate kinase)
	Apoε4	The main carrier of cholesterol in the brain with a higher oxidation of plasma Apoε4 in patients with AD
	Metal-associated proteins	Metals may play a role in the generation of ROS and Aβ aggregation; MTs, transporters, and metalloproteins are crucial in metal homeostasis
*Huntington’s disease*	HTT	Neuronal toxic protein which tends to aggregate and accumulate in neurons’ nuclei
	Oxidized proteins	An animal model expressing HD revealed a significant oxidation of α-enolase, γ-enolase, aconitase, voltage-dependent anion channel 1, heat shock protein 90, and creatine kinase
	Carbonylated proteins	Increased levels of protein carbonylation have been identified in the brains affected by HD
*Multiple sclerosis*	Carbonylated proteins	Elevated protein carbonylation has been observed in the postmortem white matter and grey matter tissue of MS patients
	LDL	Enters the parenchymal plaques of MS patients undergoing oxidative modifications within the lesion
	TTR	Abnormal protein folding due to S-sulfhydration and S-sulfonation is associated with the duration of the disease
*Amyotrophic lateral sclerosis*	TCTP	The oxidative modification of these proteins might play a crucial role in the neurodegeneration observed in ALS
	UCH-L1	
	αB-crystallin	

Abbreviations: α-syn, α-synuclein; Aβ, amyloid-β; AD, Alzheimer’s disease; CSF, cerebrospinal fluid; Cys, cysteine; HD, Huntington’s disease; HTT, huntingtin; LDL, low-density lipoprotein; LRP 1 and LRP2, low-density-lipoprotein receptor-related protein 1 and 2; MS, multiple sclerosis; MTs, metallothioneins; NFTs, neurofibrillary tangles; OS, oxidative stress; PD, Parkinson’s disease; SPs, senile plaques; TCTP, translationally controlled tumor protein; TTR, transthyretin; UCH-L1, ubiquitin carboxyl-terminal hydrolase-L1.

## 4. Applied Methods in the Study of Redox Proteomics

It is important that the methods used in redox proteomics research cover a variety of aspects, such as (i) the identification of sensors and targets that undergo oxidative modifications, (ii) the nature of the modifications in proteins, (iii) the identification of molecules that undergo oxidative modifications, (iv) the identification of specific protein oxidation sites, (v) the biological effects of respective oxidative modifications on target proteins as well as their downstream networks, and (vi) the occurrence of oxidative modifications both general and specific. In this section, we summarize the main information about redox proteomics methods (Table 2).

### 4.1. Protein Nitration

Protein Tyr nitration is widely considered a legitimate biomarker of peroxynitrite formation and irreversible protein damage, directly associated with inflammation. Different levels of 3-nitrotyrosine (3-NT) have been described and detected in biological samples such as serum, plasma, bronchoalveolar lavage, urine, and synovial fluid, among others [156]. Despite its potential as a biomarker, detecting nitrated Tyr can be difficult due to its low abundance. Additionally, 3-NT levels can lead to nitrotyrosine formation during sample processing, making it a challenging task for researchers. Two main approaches have been described for quantifying 3-NT: (a) immunochemical methods and (b) gas or liquid chromatography methods [157].

#### 4.1.1. Immunochemical Methods

Enzyme-linked immunosorbent assay (ELISA) is the most common immunochemical method for detecting specific molecules. Various assays are available, including indirect, competitive, or sandwich ELISA, which can be used for this purpose [158,159]. This method enables a high throughput at a relatively low cost and with minimal sample preparation, but at the expense of sensitivity. The use of this technique, however, may cause specificity issues, since antibodies can react with other modifications, such as chlorotyrosine or nitrophenylalanine. In addition, the accessibility of certain 3-NT residues may be limited depending on the protein sites where the antibodies bind [160].

Other methods for detecting 3-NT include immunohistochemistry or immunocytochemistry, which visualizes cellular components. Immunohistochemistry is semiquantitative and detects 3-NT in many cells and tissues. It is also used to assess the substances’ antioxidant effect by comparing the fluorescence intensity. Zhao et al. [161] evaluated brain tissue from a middle cerebral artery occlusion mouse model before and after chrysophanol administration, showing a decrease in 3-NT after treatment. A protein immunoblot analysis is widely used to measure specific proteins from biological samples, with Western blotting being a primary technique for quantifying 3-NT. Tong et al. [162] assessed the levels of 3-NT in midbrain tissue from a mouse model of PD both before and after simvastatin treatment. Their results demonstrated that the drug inhibited elevated Tyr nitration.

#### 4.1.2. Chromatographic Methods

When analyzing amino acids, high-performance liquid chromatography (HPLC) is used. The most common chromatographic method for measuring 3-NT uses UV absorption detectors at 274 nm, as described by Kaur and Halliwell [163]. HPLC-UV detects 3-NT alone or associated with proteins. However, this method has a low sensitivity and selectivity, despite its successful use in determining the content of 3-NT bound to proteins or free in biological samples [157]. Other detectors coupled with HPLC include fluorescence detection. This requires structural alterations to the 3-NT group with a derivatizing reagent since it lacks fluorescence. This modification can be achieved through the reaction of the group with 4-fluoro-7-nitrobenzo-2-oxa-l,3-diazole, as previously described [164]. This step significantly improves sensitivity and specificity. However, it also produces many fluorescent compounds, which is a major drawback of this technique. HPLC methods employing electrochemical detection (ECD) may offer a better selectivity than those using UV or fluorescence detectors. ECD is comparatively low-cost and provides an adequate sensitivity to quantify 3-NT in samples from healthy individuals, with a sensitivity up to 100-fold superior [165]. An alternative to overcoming the sensitivity issues mentioned above is to couple HPLC with mass spectrometry (MS). This method identifies different compounds and has been widely accepted as the gold standard for analyzing biological samples [157]. Other chromatographic methods described for the 3-NT analysis include gas chromatography (GC) coupled to mass spectrometry (GC-MS). However, this method requires prior derivatization and modifications to functional groups to boost thermal stability and volatility. Such modifications can lead to the formation of 3-NT, which can result in an increased standard deviation from basal levels [166].

### 4.2. Protein-Bound HNE

One extensively studied biomarker in numerous diseases is 4-HNE. To study HNE-modified proteins, a technique involving the use of solid-phase hydrazide beads (SPH) has been described [167].

This technique involves a reaction between the protein’s carbonyl group and the hydrazide group of the SPH reagent. This results in a hydrazone bond. After the reaction, proteins can be separated and purified by centrifugation. This is followed by treatment with an acidic solution to break the reversible bond formed. The main improvement of this analytical process is that carbonyl groups stay intact after hydrolysis in an acidic solution. Guo et al. [168] coupled the previous technique with liquid chromatography–mass spectrometry (LC-MS/MS) to detect and identify protein–HNE adducts in mitochondria in a rat model after exposure to HNE. More traditionally, SDS–polyacrylamide gel electrophoresis (SDS-PAGE) and Western blotting are commonly used to measure proteins [169]. Tzeng and Maier [170] used the aforementioned methods to isolate HNE-modified proteins and later labeled them with an aldehyde-reactive probe via streptavidin magnetic beads, which allowed sample enrichment.

### 4.3. Protein S-glutathionylation

In redox proteomics, protein SSG, the covalent binding of GSH to protein thiols, is an increasingly studied thiol post-translational modification. SSG regulates transcription, mitochondrial metabolism, apoptosis, and others [171]. The following are some of the methods described to detect glutathionylated proteins.

#### 4.3.1. Radiolabeling

S-glutathionylated proteins were first analyzed using the radiolabeled method [172]. For the identification of specific glutathionylated proteins in human T cell blasts exposed to OS, they used 35S labeling of intracellular GSH followed by nonreducing two-dimensional electrophoresis and MS fingerprinting [9]. In addition to its sensitivity and robustness, this technique can be employed under a variety of conditions [173]. This approach, however, has several limitations, including disturbed cell physiology caused by blocking protein synthesis. It also has nonspecific signals and cannot differentiate individual SSG sites [174].

#### 4.3.2. Mass Spectrometry

The application of MS has become one of the leading platforms for profiling redox post-transcriptional modifications on a large scale [175].

To detect the redox post-transcriptional modifications of interest, thiol reactive reagents with affinity tags (e.g., biotin) are used to selectively reduce free thiols. Since many versions of this strategy have been developed to date, it is the most widely applied method as a result of its simple chemistry, adaptability to different types of post-transcriptional modifications, and seamless compatibility with MS-based proteomics [176,177,178,179]. As a second major strategy, chemo-selective probes are used to tag specific types of redox post-transcriptional modifications for the purpose of enrichment and tagging for LC-MS measurements [180,181,182]. As a result of the distinct chemical nature of various redox post-transcriptional modifications, a variety of chemo-selective probes have been developed, some of which have been adapted for proteomic profiling [183]. Furthermore, recent studies have demonstrated the possibility of directly detecting redox post-translational modifications by MS without chemical derivatization, which had been of long-standing interest [184].

The most beneficial results are obtained when the protein solution is highly purified or homogeneous. Impurities can make mass estimation difficult. For electrospray ionization/MS, proteins must be changed to a nonionic medium since they are usually modified in an ionic environment. To determine the mass of a protein, a high initial concentration (0.2–1.0 mg) of pure protein must be used. It should be noted, however, that electrospray ionization/MS is a mild ionization technique that does not disrupt the protein–SSG bond [173].

According to a 2015 study, a quantitative proteomics technique utilizing MS was developed to profile protein–SSGs and their specific modification sites by adapting the resin-assisted enrichment method used to improve S-nitrosylation to iTRAQ reagents for isobaric labeling on resin (isobaric labels for relative and absolute quantification). With resin-assisted enrichment, noncovalent avidin–biotin binding is minimized, providing a higher specificity and sensitivity than noncovalent avidin–biotin enrichment [185]. In order to identify potential SSG-sensitive Cys redox switches, this approach was applied to RAW 264.7 macrophage cells treated with diamide and H_2_O_2_. During the study, the authors identified 364 SSG-modified Cys sites in 265 macrophage proteins that were susceptible to SSG in response to H_2_O_2_ treatment [186].

The ascorbate reduction of S-nitrosylation led to the development of Cys-BOOST, a thiol-reactive tag [187]. In addition, a cleavable Dde biotin–azide linker was introduced to enrich the tagged peptides prior to LC-MS analysis. In this Cys-BOOST approach, cells or cell extracts were treated with NO donors such as S-nitrosoglutathione or S-nitroso-N-acetyl-D,L-penicillamine to increase the number of S-nitrosylation identifications [187].

After elution with dithiothreitol, the S-persulfidation-modified proteins were collected and analyzed by LC-MS/MS. Further modifications were made by enlarging peptides by streptavidin resin after protein digestion. As a result of the elution of S-persulfidated peptides by Tris(2-carboxyethyl)phosphine, nascent free thiols could be measured using IodoTMT in a multiplexed manner [188].

#### 4.3.3. Switch Assay

In MS analysis, switch assays are particularly useful when the modification is chemically labile and unlikely to survive sample preparation [175].

Biotin switch assay is the most commonly used method. It has been widely applied to the quantitative analysis of S-nitrosylation [189] and later modified for the analysis of other oxidative Cys modifications. It comprises a versatile and robust approach for in-depth studies in redox proteomics. A biotin switch assay, which involves Grx-catalyzed reduction of SSG, was used to in situ assess the quantity of S-glutathionylated proteins in mouse tissue from various models of lung injury and fibrosis. Using fluorescent-conjugated streptavidin, biotin-labeled proteins were visualized by interacting with fluorescent-conjugated streptavidin, which demonstrated regional changes in SSG across a number of diseases [190].

#### 4.3.4. Immunoprecipitation Methods

Another method is the modified biotin switch assay, which employs the biotin/streptavidin technique to detect glutathionylated proteins. Biotin-GSH is used to label the free thiols of the newly formed protein after several steps of protecting the unoxidized thiol groups and reducing the S-glutathionylated groups. By using streptavidin beads, the modified protein can be isolated and recognized by Western blots [191,192].

Similarly, the same authors assessed another method based on immunoprecipitation of the target protein followed by electrophoresis under nonreducing conditions and a Western blot analysis with an anti-GSH antibody [192].

#### 4.3.5. Detection In Situ of S-glutathionylated Proteins

A protocol for the detection of S-glutathionylated proteins is based on glutaredoxin-1’s specificity for reducing S-glutathionylated proteins [193]. As the most extensively described and characterized glutaredoxin, Grx-1 catalyzes protein deglutathionylation. In this protocol, free thiols are blocked by alkylation with N-ethylmaleimide, followed by a Grx-1-catalyzed reduction of S-glutathionylated proteins, and the reduced Cys are then labeled with N-(3-maleimidylpropionyl) biocytin and detected with streptavidin-conjugated fluorophores [194]. In disease states caused by redox imbalances, this method may be applied in situ to determine which tissues are affected. False positives may occur due to intermolecular disulfide reduction or other oxidative modifications [194].

#### 4.3.6. Western Blot

A popular method for detecting SSG directly is by Western blotting glutathionylated proteins on one-dimensional or two-dimensional nonreducing gels with anti-GSH antibodies. S-glutathionylated proteins are most commonly identified by this method [174,195,196]. As part of a study, glutathionylation was induced using GSSG and samples were prepared for SDS/PAGE analysis. The samples were processed without reducing agents, and the proteins were resolved by SDS-PAGE without reducing agents. Electrophoresis separated samples by size and transfers them to PVDF membranes for immunoblotting with an anti-GSH antibody that detected glutathionylated proteins. An autoradiography film detected protein–antibody complexes using chemiluminescence [197].

### 4.4. Protein Carbonyls

The enormous clinical attention to these types of modifications has resulted in several types of assays to detect them [198,199,200,201]. The most extended tendency is the derivatization of protein carbonyls with 2,4-dinitrophenylhydrazine (DNPH) before analysis [202,203,204]. In a brief outline of this treatment, DNPH reacts with the carbonyl groups to form a protein carbonyl–DNPH hydrazone adduct that is easily recognizable. In turn, this trend is divided into two different approaches, immunohistochemical and photometrical-based methods. Immunohistochemical methods usually involve electrophoresis or LC, with derivatization before or after electrophoresis [205]. There are some limitations inherent to DNPH derivatization. DNPH has an affinity to carbonyl groups but also reacts with sulfenic acids. In addition, derivatization requires many hours [206].

#### 4.4.1. Immunohistochemical Detection

Immunohistochemical methods use anti-DNPH antibodies in well-known procedures such as ELISA [207] and, especially, Western blotting [202,204,208,209]. Anti-DNP assays can be used for qualitative and quantitative purposes, but also to characterize the presence and specific localization of carbonylated proteins in tissues and inside cells. Other advantages are the high specificity and sensitivity offered by immunodetection, which allows the detection of low levels of carbonyl modifications [203].

Despite this, the list of limitations is also remarkable: the binding of antibodies to DNPH unspecifically bound to DNA and other proteins in the sample, the impossibility of antibodies to access the hidden sites of the protein, and an inhomogeneity in sample preparation protocols, which lead to highly biased and not easily reproducible results [208,210,211].

#### 4.4.2. DNPH-Based Photometric Assay

The DNPH-based photometric assay is the most standardized assay for routine quantitative protein carbonyl determination. The procedure has been extensively described [212,213]. As previously mentioned, it requires a DNPH treatment. As the hydrazone adduct has a characteristic absorbance in the UV–vis spectrum, the principle driving the assay is the measurement of the DNPH absorption extinction coefficient from which the carbonyl–DNPH hydrazone groups can be quantified.

Even though recent research has tried to optimize the wide list of procedural problems of the assay [214], the DNPH-based photometric assay presents some limitations inherent to the sample. Samples containing heme proteins (e.g., hemoglobin), whose peak of absorption is close to DNPH’s, can lead to a misinterpretation of real levels of protein carbonyls [215]. Moreover, DNA and sulfonic acid (the product of the ROS-mediated oxidation of the Cys-SH group) interfere with the assay [210]. Also, a DNPH-based photometric assay requires a high number of proteins, so this test has a limited sensitivity.

#### 4.4.3. Nongel Protein Carbonyl Identification

Nongel identification of protein carbonyls follows the general approach using MS. There is another difficulty: the average abundance of carbonylated proteins in human plasma is approximately 0.2%. Therefore, it is necessary to perform additional purification steps to enrich the percentage of carbonylated proteins in the sample to be analyzed [216]. The most commonly used resource is affinity chromatography. It includes (i) biotin-hydralized derivatized carbonylated proteins/streptavidin and (ii) N’-aminooxymethyl-carbonylhydrazino D-biotin/streptavidin. Its advantage over the previous one is that while derivatization with biotin hydrazide results in an unstable product (Schiff base) that must be reduced to a C-N bond, it directly forms the C-N bond [217].

## 5. Conclusions

Cellular redox homeostasis is essential to physiological steady states. It is tightly controlled by continuous signaling to produce and eliminate electrophiles and nucleophiles. Pathological states are often associated with a shift from redox homeostasis to oxidative or reductive stress. Several antioxidant defense systems (e.g., antioxidant enzymes, GSH, protein thiols, and other antioxidants) have a vital function in maintaining redox homeostasis. The redox proteomics approach plays an integral role in detecting proteins that were oxidatively modified by 3-NT and/or HNE during neurodegeneration, thus enabling the determination of the disease-specific oxidative footprint. As the field develops technically, OS and redox signaling will be better understood as they relate to both physiological and pathological processes. For redox proteomics analyses, novel methodologies will be developed, inspiring novel material designs for treating complex pathologies in the coming years.

## Figures and Tables

**Figure 1 antioxidants-13-00127-f001:**
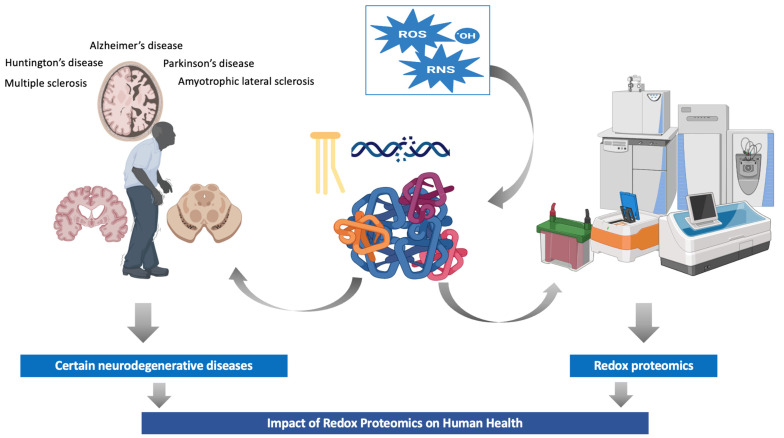
Redox proteomics: impact on human health and new insights. Abbreviations: RNS, reactive nitrogen species; ROS, reactive oxygen species.

**Figure 2 antioxidants-13-00127-f002:**
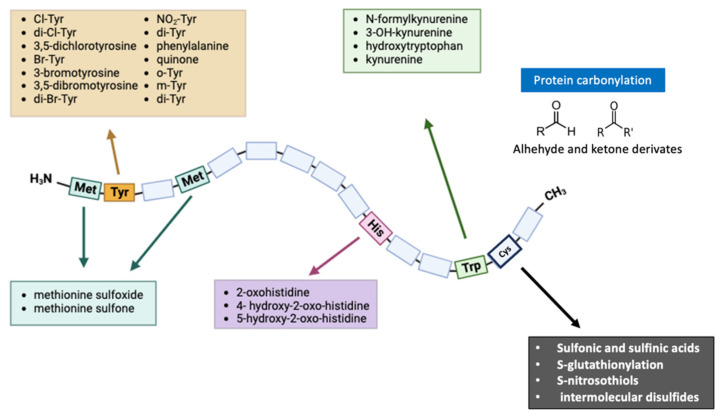
Representation of the modifications of the amino acids Cys, Met, Tyr, His, and Trp, due to lack of redox homeostasis. Various mechanisms cause an oxidative modification of several different residues in proteins. Some modifications, such as the formation of methionine sulfone/sulfoximine intermediates or the oxidation of His and Trp, are irreversible and lead to protein aggregation and degradation. On the other hand, there are reversible modifications such as the formation of methionine sulfoxides, which allow protein regulation against OS. Abbreviations: Cys, Cysteine; His, histidine; Met, methionine; Trp, tryptophan; and Tyr, tyrosine.

**Table 2 antioxidants-13-00127-t002:** Main applied methods in the study of redox proteomics.

Redox Proteomics	Employed Technique	Advantages	Disadvantages
Protein Nitration	Immunochemical methods		
	ELISA	High throughput: relatively low cost and sample preparation	Low sensitivity (i.e., accessibility to certain 3-NT) Issues of specificity (i.e., antibodies interactions)
	IHC or ICC	Semiquantitative; it allows a fluorescence intensity comparison 3-NT detection in different cells and tissues	Quantifying results is difficult Cost of the equipment needed
	WB	Primary technique for quantifying 3-NT	Time consuming, relatively expensive, sensitivity issues
	Chromatographic methods		
	HPLC-UV	Detection of 3-NT alone or associated with proteins	Low sensitivity and selectivity
	HPLC-fluorescence	Higher sensitivity and selectivity than HPLC-UV	Structural alterations of 3-NT group (derivatization) Production of other fluorescent compounds
	HPLC-ECD	Greater selectivity than UV or fluorescence detection	In general, HPLC disadvantages are the high cost and the regular maintenance
	HPLC-MS	Widely accepted as the gold standard for analyzing biological samples	
	GC-MS		Prior derivatization and modifications to functional groups
Protein-Bound HNE	SPH	Carbonyl groups stay intact after hydrolysis	May be coupled with LC-MS/MS
	SDS-PAGE (and WB)	SDS-PAGE is suitable for separating low-molecular-weight molecules	SDS causes a denaturation of proteins WB: time-consuming, expensive, sensitivity issues.
Protein S-glutathionylation	Radio Labeled	Sensitivity and robustness Used in different conditions	Disturbed cell physiology, nonspecific signals Inability to differentiate individual SSG sites
	MS	High sensitivity and specificity	Impurities, difficult mass estimation
	Switch Assay-MS	When the modification is chemically labile	
	Detection in situ	May be applied in situ	False positives (oxidative modifications)
	Western Blot	S-glutathionylated proteins are most commonly identified by this method	Time-consuming, expensive, sensitivity issues.
Protein carbonyls	Immunohistochemical detection (ELISA, WB)	Qualitative/quantitative uses, and specificity and sensitivity	Antibodies unspecifically bound to other molecules Impossibility to access the hidden sites of the protein
	DNPH-based photometric assay	The most standardized assay of total protein carbonyls	Limitations inherent to the sample (i.e., heme proteins, DNA) Very limited sensitivity
	Nongel identification	Follows the general approach using MS	Additional purification steps

Abbreviations: DNPH, 2,4-dinitrophenylhydrazine; ELISA, enzyme-linked immunosorbent assay; GC-MS, gas chromatography coupled to mass spectrometry; HPLC-ECD, high-performance liquid chromatography–electrochemical detection; HPLC-MS, high-performance liquid chromatography–mass spectrometry; HPLC-UV, high-performance liquid chromatography–ultraviolet absorption detectors; IHC or ICC, immunohistochemistry or immunocytochemistry; LC-MS/MS, liquid chromatography–mass spectrometry; MS, mass spectrometry; SDS-PAGE, SDS–polyacrylamide gel electrophoresis; SPH, solid-phase hydrazide; SSG, S-glutathionylation; WB, Western blot; 3-NT, 3-nitrotyrosine.

## Data Availability

Not applicable.

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
