# Peer review of "Using Redox Proteomics to Gain New Insights into Neurodegenerative Disease and Protein Modification"

_antioxidants, 2024, doi:10.3390/antiox13010127_

Round 1

Reviewer 1 Report (Previous Reviewer 1)

Comments and Suggestions for Authors

The manuscript, including changes and suggestions, is now better structured, clearest, and complete. 

Comments on the Quality of English Language

Some improvement in minor details about the English can yet be done. 

Author Response

Dear Antioxidants Editorial Office,

Thank you for providing us the opportunity to submit a revised version of our manuscript entitled Using redox proteomics to gain new insights into neurodegenerative disease and protein modification” to the Antioxidants.

We thank the three Reviewers for their thoughtful comments and suggestions regarding our manuscript. We have taken into account all of the comments and incorporated them into the revised manuscript in green font. An itemized point-by-point response to the comments from Reviewers is provided below in response to the changes made to the original document.

COMMENTS FROM REVIEWER 1

Comment #1: The manuscript, including changes and suggestions, is now better structured, clearest, and complete. 

Response #1: We are most grateful for the Reviewers’ highly valued comments and concerns of our study which have undoubtedly enriched and strengthened the presentation of our research in the manuscript. 

Comment #2: Some improvement in minor details about the English can yet be done. 

Response #2: Dear Reviewer, following a comprehensive review of the manuscript, certain details have been revised to enhance clarity and comprehension in the English language.

Reviewer 2 Report (New Reviewer)

Comments and Suggestions for Authors

Comments on the Quality of English Language

Author Response

Dear Antioxidants Editorial Office,

Thank you for providing us the opportunity to submit a revised version of our manuscript entitled Using redox proteomics to gain new insights into neurodegenerative disease and protein modification” to the Antioxidants.

We thank the three Reviewers for their thoughtful comments and suggestions regarding our manuscript. We have taken into account all of the comments and incorporated them into the revised manuscript in green font. An itemized point-by-point response to the comments from Reviewers is provided below in response to the changes made to the original document.

COMMENTS FROM REVIEWER 2

Comment #1: Since the title emphasizes “new sights into neurodegenerative disease”, there should be more information about why neuro is emphasized more than cardio and some brief comments on what the protein modifications are affecting to cause these pathologies, e.g., why brain may be more susceptible than some other organs and for Parkinson’s, why the substantia nigra seems to be most susceptible brain region to the redox changes.

Response #1: Thank you for the suggestion. We have revised the opening paragraph of the section titled "Redox Proteomics in Neurodegenerative Diseases" to underscore the significance of proteomics in these conditions. To maintain consistency with this approach, we have added and reorganized relevant information throughout all neurodegenerative diseases discussed in this section, as outlined in response below (please, see response #2). Comment appreciated.

Comment #2: The methods about altered protein thiols is fine; however, the connections to specific proteins and the specific protein modifications to specific neurological sites and the pathogenic mechanisms are not well developed. The review has two topics neurodegenerative disease and methods to assay redox modification of proteins and the “new sights” are cursory explanations and tangential connections.

Response #2:

Dear reviewer, in response to your recommendations aimed at delving more deeply into the associations between specific proteins, their modifications, neurological sites, and the pathogenic mechanisms of neurodegenerative diseases, we conducted an extensive literature review. This involved adding multiple references and pertinent information concerning the five described neurodegenerative diseases. Furthermore, we have introduced a new Table 2, encompassing and summarizing previously identified findings from redox proteomic studies on human neurodegenerative diseases.

Comment #3: There is an extensive list of references and many of them are not well integrated into the topic of redox proteomic influences on neurodegenerative diseases.

Response #3: Dear Reviewer, following a thorough assessment of the manuscript, we have taken the following actions: 1) eliminated references that did not align with the theme of redox proteomics in neurodegenerative diseases, and 2) incorporated additional references focusing on redox proteomics in neurodegenerative diseases.

Comment #4: A table or figure linking some specific thiol changes to particular diseases is needed.

Response #4:  Dear reviewer, in accordance with your suggestion, a new table, namely Table 2, summarizing the key findings on redox proteomics in neurodegenerative diseases, has been incorporated into the manuscript.

Comment #5: GSH is first used on Line 133 and it is not spelled out as the tripeptide glutathione

Response #5: Dear Reviewer, since the tripeptide glutathione was initially introduced in line 54, its abbreviation has been maintained without expansion in line 113.

Comment #6: Figure 2 legend. It seems “due to redox homeostasis” would be more appropriate – due to lack of redox homeostasis.

Response #6: In response to the Reviewer's suggestion, the legend in Figure 2 has been revised to enhance clarity. Comment appreciated.

Comment 7: Line 296: When first used a-synuclein should be abbreviated and then α-syn can be used thereafter. Also the gene for α -syn is SNCA (Line304).

Response #7: The suggested changes provided by the Reviewer have been incorporated as per their recommendations.

Comment #8: Gene names (PINK1 LRRK2 etc) should be italicized.

Response #8: Dear Reviewer, following a careful examination of the manuscript, all genes mentioned therein have been italicized.

Comment #9: Line 352: LRP1 is not spelled out as low density lipoprotein receptor-related protein and LRP-2 is also involved. “amyloid-β would oxidize LRP1” is not worded well, in that amyloid-β does not directly oxidize LRP1. What LRP1 does and how it affects Alzheimer’s is not well described.

Response #9: Dear Reviewer, in response to your feedback, we have taken the following actions: 1) appropriately abbreviated both LRP1 and LRP2, and 2) provided a description of the roles of both LRP1 and LRP2 in Alzheimer's disease. Your comments are highly valued.

Comment #10: Line 369: Proteins usually bind Cu, Fe, and Zn ions to control their reactivity. It would be useful to mention metallothionein here which is a major protein holding these cations and it is a protein with many Cys. And has a influence in the brain.

Response #10: The functions of metallothioneins and their role in AD have been detailed as suggested.

Comment #11: Line 386-391: The protein affecting Huntington’s disease is Huntingtin and it can be abbreviated HTT and Line 391 abnormal HTT comes from mutated HTT gene. Its first referred to as abnormal huntingtin protein and later as huntington protein. The CNS is affected by the abnormal structure of huntingtin.

Response #11: The Reviewer's recommendations regarding the accurate utilization of the huntingtin protein and its role in Huntington’s disease have been incorporated. Thank you for the valuable suggestion.

Comment #12: For MS it could be mentioned that higher number of activate microglia producing oxidants have been implicated.

Response #12: The role of microglia in MS has been detailed more thoroughly, following the Reviewer's recommendations.

Comment #13: Line 631 “glutathionylation is induced by GSSG” is not correct. Glutathionylation is R-SH + GSH to R-SSG

Response #13: Dear Reviewer, thank you for bringing this to our attention. We have rectified the description of this method. The error in the drafting, where "by" was incorrectly used, has been corrected to "using". Comment appreciated.

Reviewer 3 Report (New Reviewer)

Comments and Suggestions for Authors

The authors summarize the potential application of redox proteomics for human neurodegenerative diseases. This is an interesting topic to work on and will be significant to disease biomarker and target development. 

However, some weakness exists:

1, The authors have not presented previous achievements on redox proteomic study on human neurodegenerative diseases, especially Parkinson's diseases. The authors should perform comprehensive literature study on previous advance of redox proteomics on human neurodegenerative diseases. The authors should also provide a new table 2, which to include and summarize previous identified findings on redox proteomic studies on human neurodegenerative diseases. 

Author Response

Dear Antioxidants Editorial Office,

Thank you for providing us the opportunity to submit a revised version of our manuscript entitled Using redox proteomics to gain new insights into neurodegenerative disease and protein modification” to the Antioxidants.

We thank the three Reviewers for their thoughtful comments and suggestions regarding our manuscript. We have taken into account all of the comments and incorporated them into the revised manuscript in green font. An itemized point-by-point response to the comments from Reviewers is provided below in response to the changes made to the original document.

COMMENTS FROM REVIEWER 3

Comment #1: The authors summarize the potential application of redox proteomics for human neurodegenerative diseases. This is an interesting topic to work on and will be significant to disease biomarker and target development. However, some weakness exists.

Response #1: We are most grateful for the Reviewers’ highly valued comments and concerns of our study which have undoubtedly enriched and strengthened the presentation of our research in the manuscript.

Comment #2: The authors have not presented previous achievements on redox proteomic study on human neurodegenerative diseases, especially Parkinson's diseases. The authors should perform comprehensive literature study on previous advance of redox proteomics on human neurodegenerative diseases.

Response #2: Following the Reviewer's instructions, the role of redox proteomics in neurodegenerative diseases (i.e. Parkinson's, Alzheimer's, Huntington's, Multiple Sclerosis, and Amyotrophic Lateral Sclerosis) has been elaborated. We appreciate the Reviewer's comment.

Comment #3: The authors should also provide a new table 2, which to include and summarize previous identified findings on redox proteomic studies on human neurodegenerative diseases.

Response #3: In accordance with the Reviewer's guidance, a new Table 2 has been generated to encompass and summarize the role of redox proteomics in neurodegenerative diseases. We appreciate your suggestion.

Round 2

Reviewer 2 Report (New Reviewer)

Comments and Suggestions for Authors

No concerns

Reviewer 3 Report (New Reviewer)

Comments and Suggestions for Authors

The revised manuscript has been improved significantly. So can be accepted in the current form.

This manuscript is a resubmission of an earlier submission. The following is a list of the peer review reports and author responses from that submission.

Round 1

Reviewer 1 Report

Comments and Suggestions for Authors

This manuscript describes the main protein oxidative modification stressing its role in neurodegenerative diseases and their importance as possible biomarkers and listing the most frequent approaches used to identify them. This review is quite comprehensive and well-structured, but some improvements should be made. 

Major comments

-       The text is dense and some figures, in section 2, that show the different reactions that are described as well as tables in the next sections for example in section 4 indicating the protein modification, the different methods that can be used to analyze them and the pros and cons of each should be appreciated.

-       In section 3.1 the first paragraphs are general; I would change it to the beginning of this section or reduce it. Besides, I don´t see any specific references for Parkinson´s disease about protein redox modification as happens for the rest of the diseases that are described, and the section is only focused on mitochondrial dysfunction. Redox modification of proteins such as a-synuclein is also important in the development of Parkinson´s disease. Thus, oxidation of a-synuclein by nitration or reaction with 4HNE contributes to its aggregation which is one of the main hallmarks of this disease. 

-       Section 4.3 should also include the rest of thiol oxidations that protein can undergo and whose detection methods are quite like the ones described.       

Minor comments

-       In the abstract, I would mention, along with oxidative stress, nitrosative stress as in the rest of the text.

-       The manuscript would benefit from language polishing since some paragraphs are too complex and could be simplified and there are some grammatical errors. For example:

·      Page 2, line 79. Between instead of “among”

·      Page 6, lines 256 and 258. May destroy instead of “May result in the destruction of”. Increasing instead of “resulting in an increase in”  

·      Page 6, line 291. a-synuclein instead of “a-sinuclein”

·      Page 8, line 395. Transporters instead of “transportes”

·      Page 10, line 465. Even though instead of “In spite of the fact that”

·      Page 10, line 492. Are instead of “have been found to be”

-       The bibliography sometimes is outdated, and latest references could be found. For instance, Alzheimer´s disease incidence data is from 2015 and there are some updates (Front. Aging Neurosci., 10 October 2022. Sec. Alzheimer´s disease and related dementias. Vol 14, https://doi.org/10.3389/fnagi.2022.937486).

-       I would change the heading of section 3 because does not explain the relationship between human health and the study of redox proteome in general but specifically in neurodegenerative diseases (suggestion: Redox proteomics on neurodegenerative diseases or Redox proteomics on neurodegeneration).

-       In section 3.2, I don´t see the point in talking about DNA oxidation when the topic of this review is redox proteomics, I would remove the corresponding paragraph.

Comments on the Quality of English Language

Minor editing of English should be done.

Reviewer 2 Report

Comments and Suggestions for Authors

Comments on the Quality of English Language

Reviewer 3 Report

Comments and Suggestions for Authors

This manuscript by Cadenas-Garrido et al. focuses on the role that ROS/RNS modifications play in various human diseases.  Specifically, the researchers provide examples of how these various redox reactions disrupt cellular processes by chemically altering biomolecules.  A highlight of the review is the in-depth discussion on various proteomic methods currently being used to detect oxidative modifications in a sample.  However, there are many weaknesses in this review with many confusing aspects that will require significant edits to address.  First and foremost, there are many passages and statements of fact throughout this review that are not properly referenced using original literature.  In general, smoother segues between different sections of the review are needed to better connect and drive home the major take-away messages for each section.  The review currently reads as a list of numerous statements of scientific fact without providing enough context for the reader.  Another major concern are the lack of figures. The inclusion of new figures to break up the long passages and support the complicated pathways discussed will help to improve clarity and strengthen the specific take-home messages that the researchers want the scientific community to draw from this review (i. what are the next steps that need to be taken to address the holes in our current understanding of how ROS/RNS contribute to disease, ii. how can we improve ROS/RNS detection, and ii. where should the field move next).  This review could be significantly improved by shortening the disease section, as these have been well documented in other recent reviews.  It is also suggested that this review focus more on the chemistry behind the modifications caused by ROS/RNS and how different proteomic approaches that can be used to detect them.  By addressing these concerns, this review will become much more accessible to a broader audience interested in learning more about ways to detect oxidative stress.  Below are some other comments that are offered to help the authors improve their manuscript:

Many new figures are needed.  Please show the different ROS/RNS dependent modifications of amino acids, lipids and nucleic acids using chemical structures.  Schematics for how the different methods can be applied to detect ROS/RNS modifications would also help with clarity. 

All abbreviations and acronyms for proteins discussed in the text and figures need to be defined as soon as they are introduced in the text, tables, and figures.

Comments on the Quality of English Language

Numerous grammatical and spelling mistakes found throughout.  Please fix.